# COVID-19 Vaccination Rollout: Aspects of Hesitancy in South Africa

**DOI:** 10.3390/vaccines11020407

**Published:** 2023-02-10

**Authors:** Bent Steenberg, Andile Sokani, Nellie Myburgh, Portia Mutevedzi, Shabir A. Madhi

**Affiliations:** 1South African Medical Research Council Vaccines and Infectious Diseases Analytics Research Unit, Faculty of Health Sciences, University of the Witwatersrand, Parktown 2193, South Africa; 2National School of Government, Pretoria, Sunnyside 0001, South Africa; 3Child Health and Mortality Prevention Surveillance, Emory Global Health Institute, Emory University, Atlanta, GA 30322, USA; 4African Leadership in Vaccinology Expertise, Faculty of Health Sciences, University of the Witwatersrand, Parktown 2193, South Africa

**Keywords:** South Africa: biomedicalization, communicability, counterfactual claims, COVID-19, racialization

## Abstract

Across the globe, comprehensive COVID-19 vaccination programs have been rolled out. Naturally, it remains paramount for efficiency to ensure uptake. Hypothetical vaccine acceptability in South Africa was high prior to the availability of inoculation in August 2020—three-quarters stated intent to immunize nationally. However, 24 months on, less than one-third have finished their vaccination on a national average, and in the sprawling South Western Townships (Soweto), this figure remains troublingly low with as many as four in every five still hesitant. Medical anthropologists have recently portrayed how COVID-19’s jumbled mediatization produces a ‘field of suspicion’ casting serious doubt on authorities and vaccines through misinformation and counterfactual claims, which fuels ‘othering’ and fosters hesitancy. It follows that intent to immunize cannot be used to predict uptake. Here, we take this conceptual framework one step further and illustrate how South African context-specific factors imbricate to amplify uncertainty and fear due the productive nature of communicability, which transforms othering into racialization and exacerbates existing societal polarizations. We also encounter Africanized forms of conspiracy theories and find their narrational roots in colonization and racism. Finally, we discuss semblances with HIV and how the COVID-19 pandemic’s biomedicalization may inadvertently have led to vaccine resistance due to medical pluralism and cultural/spiritual practices endemic to the townships.

## 1. Introduction

In December 2019, a novel virus emerged causing unprecedented cases of severe acute respiratory syndrome (SARS) in Wuhan, Hubei Province, China. One month on, in January 2020, the World Health Organization designated this illness as 2019-nCoV—retitled SARS-CoV-2 in February 2020—and then onwards referred to it widely as COVID-19, coronavirus disease 2019 [1]. Worldwide, 6.6 million lives have so far been lost amid immense public healthcare crises and severe socio-economic predicaments brought on by preventive restrictions placed to curtail contagion and worsened by climate change and war in Europe [2]. Early on, the United Nations aired that the virus was “far more than a health crisis; it is affecting societies and economies at their core”—and also likely to spell disaster for this institution’s ‘Sustainable Development Goals’, especially in low-income countries [3].

Globally, governments implemented an array of socially restrictive measures to curb the spread of the virus, including stringent use of masks; regular handwashing and sanitizing; physical distancing, popularly referred to as ‘social distancing’; and, in some countries, even gender-based limitations on pedestrians sought to reduce movement [4,5]. Similarly, in South Africa, the rapid spread of COVID-19 prompted strict measures such as distancing, contact hygiene, face shields and masks, limitations on public vehicle traffic, lockdowns of public spaces, reduction of movement via curfews, and, uniquely, months-long prohibitions of tobacco and alcohol sales and transportation. While banning alcohol consumption may have been beneficial in terms of reducing lively gatherings and contagion, there was no clinical reasoning for the pause in tobacco sales, making this uniquely South African decision stand out as somewhat enigmatic. In addition, given sudden spikes in diagnosed cases, South Africa continuously extended periods of mandatory social isolation. As elsewhere, a nationwide immunization program was then gradually rolled out in February 2021 beginning with the most vulnerable.

Initially, surveys noted levels of hypothetical vaccine acceptability as high as 75 percent—meaning that three-quarters expressed acceptance towards inoculation were vaccines to become available [6,7]. Presently, 24 months into the rollout, less than 33 percent or only one-third of the South African population have finished their vaccination—two-thirds have so far opted not to immunize [2].

Reasonably, from the onset, the World Health Organization “included vaccine hesitancy in its top ten global health threats” [8]. It has thus naturally been much scrutinized in scores of studies worldwide and findings have thus far been fairly consistent with respect to the underpinnings of COVID-19 vaccine hesitancy, mentioning mainly as culprits (i) negative sentiments towards immunization stemming from misinformation obtained from largely unregulated social media platforms; (ii) counterfactual claims and conspiracy theories driving anomie and mistrust of government and health institutions; and (iii) distrust in vaccines, clinical trials, and gloomy health-risk perceptions [9,10,11]. This global corpus, however, largely ignores and neglects to illustrate COVID-19 as a social, political, economic, historical, and biological fact, which would draw out how this illness impacts on and is implicated in most dimensions of social life. Here, we dig deeper and provide one such examination from an African perspective.

In a recent study on aspects of vaccine acceptability in South Africa, we found significantly elevated levels of hesitancy and denialism in the South Western Townships (Soweto) and presented generalizable evidence that stated intent to immunize does not predict uptake (see [12]). In that same study, to explain vaccine hesitancy and denialism, we mobilized (mostly) anthropological concepts like ‘fields of suspicion’, mediatization, counterfactual claims, conspiracy theories, and othering. Here, we expand on that discussion and buttress our conceptual framework to further explore and understand social underpinnings of vaccine hesitancy—insights not just pertinent to citizens of Sowetan townships but transferable to populations everywhere. In essence, we build on our framework by drawing in elements of communicability, racialization, Africanized conspiracy theories, HIV semblance, societal polarizations, and biomedicalization.

## 2. Method

An exploratory qualitative inductive study was conducted in Soweto and Thembel-ihle, Gauteng Province, South Africa in August 2020, specifically in the township clusters of Braamfischer, Emndeni, Mapetla, Meadowlands’ zones four and five, Mofolo North, Phiri, Senaone, and Thulani townships. Soweto consists of a cluster of 29 townships and has a population of around 1.7 million people. It is populated mainly by low-income Black-African communities of predominantly Zulu, Xhosa, Pedi, Venda, Tswana, and Sotho ethnicities [8]. Ethnographic research involved interviewing in casual ambiances using semi-structured guides and writing systematic fieldnotes [9]. Qualitative data were collected by 3 senior anthropologists accompanied by 2 junior social scientists during 11 focus group discussions and 5 key informant in-depth interviews with 66 adult community members (21 men; 45 women; mean age 38; range 18–65 years). With the assistance of a local community advisory board, these were selected from the unemployed (25), employed low-income earners (15), self-employed small business owners (13), pensioners (8), and caregivers of infants or children (5). Interviews lasted from an hour to an hour and a half in the focus groups and between half an hour and an hour with key informants. These informants were purposely sampled to gather data representative and illustrative of awareness and (mis)perceptions of COVID-19 in the communities, with awareness meaning knowledge about the disease’s modes of transmission, symptoms, and vulnerabilities to it [10]. All interviewees were briefed and signed informed consent forms beforehand. The interviews and focus group discussions were digitally recorded and transcribed verbatim by an assistant. These data were then coded and thematically analyzed using computer-assisted qualitative data analysis software (NVivo12). From the content analysis, overarching themes were identified from the general areas of inquiry during the interviews, namely awareness and perceptions of COVID-19.

## 3. Results

Both the efficacy of preventive measures and the successfulness of vaccination rollouts depend heavily on whether the public receives messages, finds them credible, accepts them, and adheres to the advice of government and health authorities. Appropriate awareness about inoculation during a pandemic therefore shapes social and care-seeking behaviors. Hence, one key variable in combatting the virus is how it is mediatized by those with access to and the means of producing the dominant, overarching narrative about it. Meanwhile, dis- and misinformation is detrimental to both preventive measures and vaccine acceptancy as counterfactual claims and conspiracy theories sow doubt and uncertainty, which, in turn, generates a ‘field of suspicion’ towards immunization and authorities with levels of misperceptions resembling those seen in the early days of the HIV pandemic. With structural violence, as we shall see in Soweto, this disproportionately affects those with poor access to information and healthcare services already marginalized in a socioeconomic mire. These imbricated factors compound with certain site-specific cultural and historical contexts of the townships, which, somewhat surreptitiously, amass to amplify a field of suspicion towards COVID-19 and vaccination to the point where two-thirds the population (⅔) have been swayed into hesitancy and denialism due to confoundment, incertitude, and doubt, leading to adoptions of ‘alternative forms of knowledge’ (counterfactual narratives). This then triggers a social mechanism medical anthropologists refer to as ‘othering’, which, as we found and illustrate here, evolved into more sinister shapes in Soweto, namely those of racialization, Africanized conspiracy theories, and societal polarizations due to the reproducible nature of communicability and a strict biomedicalization of COVID-19 where traditional medicine is widely used. These are highly detrimental to vaccine confidence and fuel hesitancy considerably and, consequently, must be addressed to boost uptake in present immunization programs. As such, appropriate mediazation, alongside a debunking of counterfactual claims, is pivotal in driving forward inoculation.

## 4. Discussion

### 4.1. Mediatization

Statements recorded in Soweto regarding clinical knowledge about COVID-19’s modes of transmission, symptoms, vulnerable groups, and actions in event of illness generally correlated with what might be characterized as relatively good awareness, even with respect to new variants. This echoes findings of other studies undertaken throughout South Africa [13,14] and elsewhere, which concomitantly accentuate “the importance of knowledge and awareness of the COVID-19 vaccine to increase acceptance rates” [15].

Regarding modes of transmission, most community members knew that people may contract COVID-19 by way of touching, proximity, airborne saliva droplets, handshakes, or not wearing a mask.


*If someone with COVID touches a paper, and I come and touch the same paper without disinfecting it, I expose myself to the virus, and I can be infected. (Focus group two)*



*You can get infected if you’re in a crowded place without wearing a mask and not practicing social distancing. (Focus group eight)*



*[Through] shaking hands and by coughing—when you cough without covering your mouth. (Key informant four).*



*We’ve been told to always wear a mask and wash our hands, because it’s passed on through coughing; those droplets can be passed on to the next person as well as their sneeze. (Focus group nine)*


Sowetans described COVID-19 symptoms as cold spells, fever, weakness, fatigue, and a loss of taste or sense of smell, though some symptoms were not mentioned at all, including myalgia, malaise, nasal congestion, expectoration, rhinorrhea, headaches, or nausea (reported in [13,16,17,18]).


*I mean, the symptoms are flu, the throat feels tight, like you’re developing asthma. (Key informant one)*



*The patient will be coughing; some will experience shortness of breath. (Key informant four)*


Of course, some groups are more vulnerable than others, including “racial/ethnic minorities, children, the elderly, immigrants/refugees, those who are socioeconomically disadvantaged, disabled, underinsured, from rural communities, incarcerated, facing domestic violence, LGBTQ+, and [those] with certain medical conditions.” [19,20,21]. Again, correctly, informants pointed mainly to children, the elderly, healthcare workers, and those with chronic illnesses as vulnerable. Folks in one focus group pensively nodded and largely agreed that the most vulnerable were people with chronic illnesses.


*My mum, because she’s diabetic, and they said that people who’re diabetic or sick have weak immune systems, which puts them at a higher risk of catching the virus. A person with a chronic disease is one with diabetes, high blood [pressure], HIV, and those who have cancer. It’s anything that’s chronic in your blood. (Focus group three)*



*If it [COVID-19] finds you in good health and your immune system is strong, that’s when it leaves you alone, but if you’re unhealthy, that’s when it can even kill you within three days. That’s the challenge we have with this new virus. (Focus group nine)*


One notion that added considerably to anxiety, however, was that an individual may be infected with COVID-19 and remain asymptomatic while contagious.


*The government said people can contract the virus and show no signs but can pass it on to the next person. (Focus group six)*



*At times, you can be positive for COVID, but not see those symptoms. (Focus group eight)*


Asymptomatic patients are indeed hard to trace, unlikely to self-isolate, and more likely to retain normal social patterns, making this preoccupation comprehensible [22]. With respect to self-isolation and quarantine, most everyone was up to speed.


*I know that when you’ve contracted it, then you have to quarantine—when it’s full at the hospital, take care of yourself, and practice social distancing until you recover. (Focus group five)*


It soon became clear that community members were eager to keep abreast of developments and information regarding the pandemic. Some accentuated how they kept themselves updated through largely unregulated social media.


*[People] know about the virus, they are aware of it through the media, social media. (Focus group nine)*


Regarding mediatization, as Briggs points out, people do not exist in societies with media, but rather in “mediated societies where images of self and society are shaped by the media.” [23] The public is then very much molded by communication because public discourses produce the public they pretend to address as people “position themselves in relationship to circulating messages” (ibid.). Ideally, the dominant public narrative, here about COVID-19, would be produced by those with access to production of authoritative knowledge about disease. In South Africa, one would assume this power to lie with the “primary definers of emerging narratives about the pandemic”, namely public health experts and policymakers (ibid). These, in terms of generating literacy with respect to purely clinical aspects of COVID-19—transmission, symptoms, quarantine, et cetera—seem to have been successful in this mediatization, as we saw above. However, contrarywise, regarding the nature, origin, and general perception of COVID-19, parallel and fraught discussions ensued about “COVID-19 cover-ups, pandemic geopolitics, bot and humanly driven disinformation floods on social media, and reporting bias in the press” [24].

For this, Fairhead et al. employs a term, ‘fields of suspicion’, in which the decision to inoculate is taken within a field of “uncertainty and speculation in which wider confidence or worries are relevant” [25]. For Marsland, this suspicion and uncertainty (or ‘not knowing’) is a combination of “secrecy, uncertainty, and skepticism in related medical knowledge”, especially where there are conflicting messages from formal authorities alongside flurries of unsubstantiated rumors breaking hourly on unregulated platforms such as WhatsApp [26]. So, whereas clinical dimensions of COVID-19 may have been effectively mediatized, unfounded discourses and counterfactual claims were widely propagated qualified by nothing but the likes of YouTube videos, ‘tweets’, or unfounded claims on social media and messaging services, thus dramatically amplifying a field of suspicion in the sprawling townships.

### 4.2. Communicability

In Soweto, early on, some mentioned that they had been following COVID-19 on television and seen events unfold in other countries:


*My family was very frightened. At first, we didn’t know what this COVID-19 is, but from watching the news—Italy, China—people are dying, and then we’re like: ‘no, people are dying; we are not going to go out [of the house]’. (Focus group eight)*


Some community members were aware of the presence of COVID-19, but due to deep uncertainties about it, this awareness fostered only fear.


*I’m still very scared of Coronavirus, even today, because I’ve not yet understood what it is or when it’ll be cured. (Focus group nine)*



*There’re still people who don’t understand anything about COVID, yes, they know it kills, but they don’t where it’s from or how it arrived. (Focus group six)*


In contemplating these anxious statements, firstly, we should be mindful of the townships’ levels of access to formal media and, secondly, of barriers to comprehending health-related messages caused by the plethora of languages spoken throughout Soweto.


*No, in my community, there aren’t a lot of people who have access to televisions. (Focus group six)*



*I’m 99 percent sure that in our community, our people, there is a communication barrier—people from Malawi, people from Zimbabwe, people from Maputo [Mozambique], others from other countries, they come. How brave they are! They come to South Africa, no visa, no nothing, just coming. Now when you try to initiate communication, they just smile. So, how are you going to give that person information? So, it becomes a problem. (Key informant four)*


‘Ignorance’, as one youth characterized it, or ‘lacking interest’, were also touched upon as hurdles to knowledge acquisition. One man’s experience was that:


*We had pamphlets that were being distributed, but people were taking those papers, and would not even read [them]. You’d find them on the floor, and you know, sometimes, you’d ask a person: ‘what does this thing say’, and they would respond saying: ‘hay, I dunno, I just saw it here.’ So, ignorance and misinformation are the most important reasons why people fear this disease. (Focus group seven)*


So, once an infectious disease appears, health surveillance and control systems are activated, which includes the propagation of relevant and accurate information to the public to prevent further infection and ensure that appropriate preventative behaviors are adopted and practiced [27,28,29]. South African government and health authorities actively mediatized COVID-19 messages through various outlets and community members were made well aware of clinical aspects of the disease. Meanwhile, due in part to poor access to formal media or a lack of interest—but owing hugely to COVID-19’s multidirectional and jumbled mediatization on Internet-based media with little or no regulation (and a hence growing field of suspicion towards immunization and authorities in general)—most people found themselves deeply anxious, mistrustful, and full of uncertainty and doubt in the face of a deadly disease. This leads us to communicability.

As touched upon while discussing mediatization, the wavering narratives we recorded in Soweto are rooted in the production and reception of knowledge about disease, meaning in its mediatization or ‘construction’ of public discourse. This communicative process is ideologically constructed in such a way “as to make some people seem like producers of knowledge, others like translators and disseminators, others like receivers, and some simply out of the game” [23]. For this, Briggs proposed the term ‘communicability’ to draw attention to its productive capacity, which is to say that we need to ponder not just the content of messages but “how the ideological construction of their production, circulation, and reception shapes identities and social groups and order them hieratically” [23]. As such, this productive capacity of communicability buttresses other power-laden processes such as racialization and biomedicalization, as we shall soon demonstrate, when the fundamental human ‘will to truth’ is (mis)led by uncertainty and doubt, yet the “erosion of trust in authoritative and scientific knowledge does not extinguish the ‘will to truth’”, but rather “opens a space for alternative forms of knowledge”—counterfactual claims and conspiracy theories about the nature and genuine origins of COVID-19 [24].

### 4.3. Counterfactual Claims

Counterfactual claims have certain shared aspects—ill intentions are implied, ‘something is off’, and there is a (hidden) agenda or motive as to why this or that is happening. Moreover, conspiracy theories are solipsistic and immune to evidence and “inherently self-sealing—evidence that counters a theory is reinterpreted as originating from the conspiracy theory” [30]. Meanwhile, theorists, here ‘anti-vaxers’ or vaccine denialists, see themselves as victims of persecution and, at the same time, self-perceive as ‘good truth-seekers’ unravelling a wicked conspiracy.

In these alternative realities, COVID-19 fatalities are exaggerated on purpose; it was a scheme concocted by ‘Big Pharma’, or rather it was thought up by shadowy entities of the ‘deep state’, while some believe it does not exist at all. In other iterations, the United States’ army covertly brought it into China, whereas some have it that it escaped from a laboratory there, or that Bill Gates put microchips in it to ‘control’ people, or that the 5G network caused it. Others again claim that COVID-19 is in fact a biological weapon run amok or that the virus sprung from genetically modified crops. In Soweto, we discovered several local or ‘Africanized’ counterfactual claims.

When such misinformation abounds, alongside uncertainty and doubt in a growing field of suspicion, does it not become more understandable why so many might hesitate and postpone immunization, perhaps indefinitely?


*When COVID arrived, I saw a lot of things—the way people were ignoring COVID. Because when COVID arrived, there was lots of false information circulating through the social networks. (Focus group one)*


### 4.4. Othering and Racialization

In the early days of the COVID-19 outbreak in South Africa, despite government and health authorities airing newsflash after newsflash monitoring its rapid spread across the country, our informants openly questioned whether the virus was real.

*In my location [*kasi*; neighborhood], there’re many people who don’t believe that the virus is real. They only believed it when many people started dying and by then the virus had already spread everywhere. In Mofolo, they don’t believe COVID is real. They’ll tell you that it doesn’t exist. (Focus group two)*

With beaming round spectacles, one thoughtful man speculated that a relatively low number of fatalities in the communities was contributing to people questioning the disease’s existence.


*Only few people died in our community, and this made people think that the virus isn’t real. (Focus group three)*



*I also used to say that this thing doesn’t exist, more specifically when it started, and they showed other countries. I just assumed that it was something far off because no one knew anyone who was infected at the time. (Focus group six)*


That the COVID-19 outbreak began in Wuhan, China in December 2019 at the Hunan seafood market soon became a well-established fact among informants. However, the presence of unsubstantiated rumors and doubts as to exactly how the virus came about in China led to scapegoating and stigmatization of those of Asian descent as an outcome of fear, blame, and finger-pointing visible at the time as increased hate crimes against Asians [31]. One common speculation was that the virus had arisen due to consumption of certain foods—foods eaten in China, more precisely.


*Animals?! That’s what many say. I even saw it in a movie that the virus came from animals. (Focus group three)*



*Is it true that the Chinese got the virus because they eat dogs? (Focus group two)*



*They, the Chinese eat everything, snakes. (Focus group three)*



*They said it originates from China, they ate pigs, and some ate frogs. (Focus group five)*


First, notice above the underscored ‘they’ (as an opposition to ‘us’). Here, we see a social process that medical anthropologists have labeled as ‘othering’ [32]. As a behavioral mechanism, it is an affirmation of belonging and assertation of social identity in the face of uncertainty and angst in which ‘they’ (the Chinese) become vile antagonists with toxic, repulsive foods and unsanitary customs, while ‘we’ (‘us’) have done no wrong. Onoma described this process of constructing others as public health dangers as ‘scapegoating’ and demonstrated how this reflects “a global pandemic of xenophobia that accompanied the public health crisis of COVID-19” [33]. In Soweto, however, we found ‘othering’ to morph into a more sinister shape, namely that of racialization as far-reaching effects of associating COVID-19 with Chinese nationality in the townships became clear [31].


*I think the reason why Corona is growing is because of denial—believing that there’s no such thing as Corona and that it won’t kill you. That’s what they say in the townships: ‘it’s the Chinese disease’. (Focus group three)*



*Yes, the rumors are many. At first, I was afraid. At China Mall (in Johannesburg) even now, if you’re climbing into a taxi, they say you’re going to ‘Corona City’. I was afraid. I only go, because I have to, but I’m afraid of that city. (Focus group three)*



*It is the Chinese fighting over the 5G. We heard those stories—Trump and China. (Focus group three)*


In a slur, Donald Trump, then democratically elected president of the United States of America, referred to COVID-19 at the time as the ‘Kung-Flu’—blatant racialization, which has been defined as “the extension of racial meaning to a previously unclassified relationship, social practice, or group” [23]. Problematically, racialized populations “are deprived of agency” and become “marked in ways that can long outlive epidemics” (ibid.). At the same time, counterfactual claims and conspiracy theories referenced colonialism, militarization, and racialization throughout regions [34]. As such, the COVID-19 pandemic “loaded onto already existing socioeconomic inequalities, racial discrimination, and uneven access to healthcare” [35]. For that reason, we encountered several Africanized variants of racialization due to region-specific histories of colonialism, racism, and social inequality. For instance, one motive to deny susceptibility to COVID-19 was the assumption that Black Africans are somehow not vulnerable at all due to ‘culture’ and ‘strength’.


*Other people were sharing cigarettes and traditional beer, saying: ‘hay, that thing doesn’t exist’. ‘I am Tsonga and I believe in my culture; it will not catch me’. (Focus group five)*


*They would tell you that: ‘we’re not the same as other countries; we’re eating the foods that we want, we eat *masonja* [mopane worms], we eat *gushe* [samp], meaning ‘I’m strong, because I’m from Limpopo’. (Focus group five)*

Gesturing with her hands, a spirited woman underscored the belief that the genetic make-up of Black Africans would shield them from infection.


*We have the mentality that, we as Black people, our immune system is strong. They’d mention that we as Black people have strong genes and our immune system is strong. (Focus group one)*


This instance of racialization can be found throughout Africa, involving notions that “African blood is stronger” or “our blood is of higher quality than that of White people” [25]. A girlish youth looked around from face to face in the group and then asserted her logical conclusion that:


*Zulu people say that COVID only infects Whites, because they are weak, but not them, because they are extraordinarily strong. (Focus group one)*



*It only killed White people and we as the other races are very safe. (Focus group five)*


Connecting racialization and medicalization through communicability “reveals how their power is derived partly from ideologies of communication with which notions of race and health are imbricated” [23]. The Whites are ‘weak’, the Zulu are ‘extraordinarily strong’, and it ‘only killed White people’, not ‘us’, more othering, progressed into racialization. When a pandemic is racialized, health professionals and reporters “place the production and circulation of knowledge about race within biomedical spheres of communicability”, which is why biomedicalization might itself become part of the problem, as we shall see.

### 4.5. Africanized Conspiracies

As illustrated, most Sowetans believed COVID-19 to somehow ‘come from China’. A grizzled woman, visibly unsettled by the topic, concurred with this sentiment, but added yet another preoccupation regarding the pandemic’s origins.


*I believed that it was something manufactured by them [the Chinese]. Us? We know that it’s man-made! (Focus group five)*


This exact view was repeated endlessly.


*Some will say that’s something that has been concocted by White people in the air; they want to kill us! The mere fact that it’s taking them this long to find a—I’m not sure—a cure. Yeah, that’s the problem. I think it’s man-made. (Focus group eight)*


Here, again, ‘the other’ (Whites) has put ‘something’ in the air and ‘they want to kill us’ (Blacks)—othering transitioned into racialization wrapped up in counterfactual claims of the virus being ‘man-made’. For a fully-fledged African conspiracy, all we need now is to pile on nefarious intent, preferably with underlying malicious motives. Others in that same focus group engaged and provided those motives, echoing that:


*The population is too much [too numerous]; that’s why they’re trying to reduce it. (Focus group eight)*



*They’re trying to start from scratch. It is a … chemical [biological] bomb! (Focus group eight)*


Eventually, an African extinction agenda was plainly alluded to, specifically in comments concerning a certain conspiracy to annihilate (Black) Africans and how African leaders were responding to it.


*There’s a vaccine that’s being distributed throughout Africa and you Africans shouldn’t take that vaccine; they’re trying to kill you. There was this African president, I don’t remember the name, he was saying we shouldn’t take the vaccine. (Focus group six)*


Amid rumors of the virus having been purposively manufactured by ‘man’, there were often whispers and propositions that the purpose of ‘concocting’ the virus was really to control a population boom. For instance, one mother cradling an infant worried that:


*There was a rumor saying they want to half the population of the [Black African] people. (Focus group one)*


Some of our participants genuinely believed there to be a deliberate effort to control Africa’s Black population by way of ‘testing’ a vaccine on it.


*There was also a rumor about the guy that funds [our hospital]—Bill Gates—he wants to test his vaccine on Black people only. (Focus group one)*


Once more, thinking about fields of suspicion towards inoculation and authorities, can there be any doubt by now what an amplification of this smoldering field might mean when fueled by scenarios in which White people underhandedly stage a viral genocide against Black Africans? The answer and outcome, of course, is vaccination hesitancy and surely a rise in expressed anxieties. This, however, is nothing new, and it is here worth noting certain semblances between the COVID-19 and HIV pandemics. Over four decades into the HIV pandemic, since its first reported case in 1981, misconceptions about the disease stubbornly persist to this day, as portrayed by Mwamwenda, including claims that:

*HIV is a disease of Black people; HIV was the creation of people who wanted to exterminate Black people, such as Africans, African Americans, and homosexuals; God sent HIV as a means of curbing or destroying sexual immorality; and HIV is transmitted through mosquitoes* [36].

In the face of similar (nonfactual) claims about COVID-19, certain community members appeared to adopt diverging narratives (or ‘alternative forms of knowledge’) as a mean of resisting authorities in a manifestation of narratively constructed vaccination hesitancy, including claims along the lines of “people believed that the virus cannot thrive in Africa due to its hot climate and thus felt immune to infection”; or that the disease is “meant for the rich and politicians” [37]. Again, this form of recusation and denial was often seen during the early days of the HIV pandemic, as prominent leaders and politicians persistently maintained that there were no clinical proof or link between HIV/AIDS, contrary to scientific evidence [38,39].

### 4.6. Societal Polarizations

COVID-19 also mirrors certain aspects of the HIV crisis as people—afraid and distrustful—turn to racialization and deny that they themselves should be at risk as an outcome of a social identity-related schism, i.e., the transformative othering between the (possibly HIV) ‘positive’ and the ‘negative’, between the ‘Whites’ and the ‘Blacks’, between the ‘patient’ and the ‘person’, as Steenberg has described [40]. Mind that with HIV, this exact same psychological function plays out today, though it has long been established beyond any reasonable doubt who it infects the most; where it came from; whether it was man-made or not; and whether it was ‘manufactured’ to control or reduce specific ethnic populations—which of course achieves little but a deeper sowing of doubt and incertitude.

While several informants associated COVID-19 directly with White people, they did not necessarily share the opinion that Black Africans should be exclusively immune to the virus. Instead, the defining othering was made on socioeconomic grounds.


*There was also the rumor that the virus only infects rich people and those who travel overseas, but the poor were immune. (Focus group two)*



*We believe that COVID is for people who have money. In the beginning, saying: ‘it’s a White man’s issue, rich men’s disease, people that are flying’. We’re not flying to the States, we’re just ordinary people staying in the township, hence when it first impacted the Western Cape, we said: ‘yah, that’s a place for rich people’. (Focus group seven)*



*I used to say: ‘it only affects those who have money’, because none of our families were affected. At the time it started, we were told only White people and people who have money, who travel, can contract it. People at the hostel [autocratic residential compounds] would say: ‘it’s for people who travel abroad, so since they don’t go anywhere, they can’t have it’. It was meant for the wealthy people, the people that were dying, most were people from Italy. (Focus group six)*


Comparable claims have been reported from elsewhere in Africa, where views included that the virus was a “rich man’s disease” and cannot infect the poor [41]. Aminu was told that COVID-19 is a “White people disease”; an “imported disease”; and that it is a “disease of the wealthy people” (ibid.). Such speculation “interplays with wider worry about ‘White people’ and their wealth more generally”, which then relates not only to race but now also to issues of distribution of power and wealth [25]. We can say with Manderson and colleagues: “As their lives unfolded during the [COVID-19] pandemic, it became clear that the unequal distribution of power and wealth across class, gender, age, and location created and perpetuated power inequalities and inequities” [35]. Thus, with structural violence, ways in which counterfactual narratives are communicated about COVID-19 produces not only a rise in expressed anxieties and magnified (fields of) suspicion—which sway people into vaccine hesistancy and denialism—they also exacerbate existing polarizations between race, class, and location, and deepen prevailing estrangements: the White from the Black, the rich from the poor, the one ‘flying to the States’ from the *Mkhukhu* (shack) tenant in Soweto.

As we saw, the pandemic’s mediatization, at least as broadcast by health authorities and politicians, was placed within biomedical spheres of communicability. However, this approach “medicalizes politics because it sidesteps questions about the distribution of wealth and discrimination based on ‘race’” [26]. The prominence of such power inequalities and societal polarizations as reoccurring themes in counterfactual claims and conspiracy theories is a strong indicator that these are expressions of resistance towards existing authority and can in fact “index deep-seated social-structural dilemmas or group-specific, locally recognized legacies of maltreatment such as those linked to racism or colonialism” [42].

### 4.7. Biomedicalization

The stringent biomedicalization of COVID-19 and unilaterally clinical mediatization from health authorities may have augmented resistance towards those same authorities in the townships due to existing medical pluralisms in Soweto, mainly traditional phytomedicines (*muthi*) and healers (*sangomas*)—cultural and spiritual practices endemic to the townships. Others, in turn, swear to faith healing or allopathic medicines. Several informants reported to have acquittances in their communities who repelled the idea of going to a hospital due to a public perception of hospitals as ‘risky places’.


*In our townships, a lot of people die when they go to the hospital, so our mindset is that the symptoms get worse once you go to the hospital. (Focus group three)*


Whereas most informants were not entirely impervious to the idea of visiting a hospital for COVID-19 treatment, they did, however, bemoan that hospitals would not provide them with indigenous remedies.

*At the hospital, they don’t allow *umhlonyane* (wormseed; traditional medicine) and garlic. The hospital just gives you hot water. (Focus group three)*


*I believe it’s better to self-isolate at home because there it’ll be manageable. There was a rumor that if you drink artemisia [wormseed], it assists, so when you get to the public hospital—and that’s the place the government gives you to isolate—there’s no access to garlic and stuff. So, it’s better to be at home. (Focus group three)*



*But they also make garlic water and lemon. There was a company where 32 of their employees had COVID. 11 of them got admitted to the hospital and they gave them these things; they all healed. (Focus group three)*



*Artemisia has proven to be more practical and artemisia is more of a cultural thing. (Focus group three)*


In addition to traditional medicines, informants related that some people in the communities had come up with an alternative remedy to ward off COVID-19.


*My customers would say: ‘we sanitize with alcohol’ [laughs]. Yah! [It means that] you have sanitized the insides [giggles]. There’s another uncle, who says that if he drinks alcohol, the hot stuff, he won’t get sick, because the alcohol percentage is high, and if sanitizer also has alcohol, that means he’ll be fine. (Key informant one)*



*We believe that the more alcohol you drink, the more this virus won’t get you. (Focus group one)*



*Yes, I heard we should drink it [alcohol] from time to time because it helps. (Focus group three)*



*My cousin is always with a Gordon’s [gin]. He encourages people to drink it because it helps the lungs. (Focus group three)*


Some appeared enticed and willing to resort to more extreme measures to neutralize the virus in what at times seemed like barely concealed substance abuse.


*Someone even said that we should drink the sanitizer. (Focus group two)*


In addition, informants oftentimes possessed indigenous knowledge about how to traditionally ameliorate certain symptoms that present like those of COVID-19, meaning that this disease may, however inadvertently, be treated as another condition altogether—and by traditional means. Again, nothing novel, some communities have long been known to believe that HIV can be cured by traditional healing. Mphahlele have also shown people to believe that COVID-19 is curable by means of traditional medicines listing indigenous herbs, such as eucalyptus-infused steam, which people have been using alongside alcohol as curative measures throughout the pandemic [43].

Models of (bio)medical causation (aetiology) constitute ways of thinking about the world and interacting with it. Biomedicalized narratives about epidemics therefore “make racial and sexual inequalities seem natural as if bacteria and viruses gravitate toward populations and respect social boundaries” [23]. As such, as we have illustrated here, COVID-19 is a social, political, and biological fact, and its examination is “a launching pad to draw out how any disease is implicated in and might impact on social life” [35].

## 5. Conclusions

Though initially ascertaining peoples’ clinical well-awareness of COVID-19, in shook township communities, we descended into a maddening maze of false binaries—Africanized, racialized, and socially polarizing counterfactual claims built upon towering foundations of global conspiracy theories propagating everything in between a non-existence of COVID-19 and it being man-made—covertly designed to decimate populations and races. Others, due to medical pluralism and (resistance towards) the biomedicalization of the disease, simply turned to herbs and alcohol. This potent blend of facing life-threatening illness amid officially mandated limitations on freedom, nagging doubt about what is going on ‘behind the stage’, mixed up with covered-up social angst behind masks and daily casualty reports, transforms into and manifests as nervous hesitation to immunize. Counter- and nonfactual claims about COVID-19 may stem from misinformation, which then in turn results in variegated nuances of hesitancy, but we also discerned how these are truly expressions of resistance towards official doctrine and authority stemming from existing power inequalities that further aggravate societal polarizations during a pandemic. Growing fields of suspicion were fueled by fear and doubt, which, again, led to othering and its evil twin, racialization, the ‘Kung-Flu’. In Soweto, this was exacerbated by lacking access to formal media and (multiple) language barriers but, most importantly, by lingering site-specific structural violence brought on by legacies of colonialism and apartheid, fostering further resistance towards proponents of inoculation and officially blue-stamped health-related discourses. As such, understanding COVID-19 vaccine hesitancy in South Africa is—as much as it has been and still is the case with HIV—as much a matter of exploring its social aspects as it is of addressing its clinical dimensions, lessons which are best remembered during future outbreaks. It follows that it remains paramount during vaccination rollouts to unveil and address aspects detrimental to vaccine confidence and selectivity, especially in lower-income groups for underlying, context-specific cultural, spiritual, historical, and socioeconomic reasons. Appropriate mediazation alongside a debunking of counterfactual claims is crucial in driving forward immunization.

## Data Availability

The data presented in this study may be available on request from the corresponding author. Restrictions apply to the availability of these data. The data are not publicly available for ethical reasons.

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
