# Peer review of "COVID-19 Vaccination Rollout: Aspects of Hesitancy in South Africa"

_vaccines, 2023, doi:10.3390/vaccines11020407_

Round 1

Reviewer 1 Report

Dear Authors, thank you for the opportunity to review the manuscript. Attached you will find my comments:

This study aims to explore aspects of Covid-19 vaccination hesitancy in South Africa and couls reveal some very important informations for the developement of future vaccination campaigns in South Africa.
Community members of several South African provinces were interviewed in an exploratory inductive qualitative study on overarching themes related to awareness and perceptions of Covid-19 within focus groups and in-depth interviews.
The results show that vaccine hesitancy towards COVID-19 in South Africa is influenced by social aspects, as well as lack of access to media, language barriers, ongoing site-specific structural violence and consequences of the legacy of colonialism in the past.

I would recommend: ‘minor revision’.

Abstract

Overall comment: please structure your abstract by using the established categories: introduction, methods, reults, discussion. 

Line 10-12: Please indicate source for this data: how do you know about the intention to get vaccinated in August 2020? Were there any public surveys?

Line 9-24: Please provide brief information on methodology.

Line 17: Othering please put in inverted commas and explain meaning.

Manuscript

Introduction:

Line 33: „6.6 million lives have so far been lost worldwide (…)”. Please indicate the source of these data.

Line 47: Why were sales and transportation tobacco and alcohol important measures in Africa? For example, are they important sources of income for the population? Please specify.

Line 51-55: How exactly were completed vaccinations measured? Were vaccinations refused or were there other reasons (access, availability, prioritization, health policy measures etc.)?

Line 58: please remove a blank between the both words “cannot” and “predict.”

Line 62-63: the findings are not really transferable to populations everywhere…please refer better to the context of your research project.

Line 63-65: Really comparable when it comes to vaccination hesitancy and there is no vaccination option for HIV?

Method:

Line 79: “These informants were purposely sampled to gather data representative (…).”  Does that make it truly representative?

Line 84: What issues were identified in relation to awareness and perception of COVID-19? Please specify the themes.

Please explain the method you used to analyze the data: did you apply a qualitative content analysis.

Findings:

Line 92-96: Where do these conclusions come from? Please state how you arrived at this conclusion.

Mediatization:

Line 117: “If someone“ instead of  “I someone”.

Line 117: For the overview, please indicate direct citations clearly and specify the relevant participant- age, gender, focus group or in-depth interview, state residence/ community.

Line 120/ 125: you indicate that quotes come from “focus group eight/ focus group nine”. In Line 75 (Methods) it is written, that there were “four focus groups” all together. Please specify.

Line 165: Please remove the explicit naming of the author (“Briggs”), as this disturbs the flow of reading. Above all, if the author is not known to the reader, there is no added value in direct naming.

Othering and Racialization:

Line 267: please correct the spelling “Othering and Racialization” instead of “Ohtering and Racialization”.

Line 283-287: Please substantiate statement with sources.

Line 302: “Othering” please put in inverted commas.

Conclusion:

Line 519-524: Can you really conclude that from the results? Please refer more to your results.

Line 528: What kind of  practical recommendations or implications for practice can be derived for the entire population?

Thank you four your interesting contribution!

Best regards

Reviewer 2 Report

Thanks a lot for the chance to review this manuscript. Steenberg et.al. did a qualitative analysis for COVID-19 hesitancy among mainly low income populations in South Africa. This is very interesting work. 

I only have two minor comments:

 1- Could you please make the sentences and quotations taken from the participants in italic? It will help the reader to easily find them. 

2- Although the Results (or the findings) and the discussion were merged together (which is fine). I could not see a good comparisons with other finings from similar studies, especially vulnerable populations. It will be good to compare you results. This is an example reference  DOI: 10.3390/vaccines10101634

Reviewer 3 Report

Vaccine

Vaccine 2050124

Comments to Authors

COVID remains a serious global health threat. Despite the wide availability of effective vaccines, large portions of the population continues to be unvaccinated and hesitant to do so in the future. Your work to understand an especially high vaccine hesitancy rate in South Africa is very valuable to developing strategies to mediate the impact of COVID on vulnerable populations. I have offered some comments and suggestions that I hope you will find useful in your current and future work in this area.

Introduction: COVID vaccine hesitancy has been studied throughout the world and the general findings have been fairly consistent regarding the impact of various conspiracy theories and the spread of non-factual information regarding vaccines. While there may be some unique features of the South African population you are studying, I believe a summary of the general literature in this area can be very useful and should be presented or, at least, acknowledged.

Methods:  You do not indicate whether this study was reviewed by an ethics board. There should be some indication that this research was submitted and approved by some REC.

There needs to be more information about the focus groups and interviews available to readers. Who conducted the data collection? How long were the focus groups and interviews? Was a general outline used to ask questions and gather data? Describe the general areas of inquiry for the focus groups and interviews.

Findings: The first paragraph in this section does not reflect any findings or information that should be included in a presentation of study results. It belongs in the Introduction or, perhaps, in the discussion.

Mediatization: The last two paragraphs seem to be a narrative summary of the theory and literature regarding mediatization. Again, this has no place in the presentation of findings.

It appears that your only findings in this area were that participants had knowledge of COVID that was consistent with that of public health experts. Were there no contrary beliefs?

Communicability: I don’t understand how you make the argument that information is not being communicated to the population when in the previous section you indicated that the participants seemed to have a good understanding of the virus, how it is contracted, and how to mediate its spread. Furthermore, you suggest that there was a lot of suspicious on the part of the population but you present no data to support that claim.

The last three paragraphs, as in the previous section, is not a presentation of data. It does not belong in the Findings section of this paper.

Counterfactual Claims: Most of this section is not a presentation of data. There is only one brief quote to vaguely describe “false information.” If more specific claims, assertions, and beliefs exist, it should be presented from the data.

Othering and Racialization: There is quite a bit of this section that is beyond the scope of a straight-forward reporting of the findings. The interpretation of the data should be presented in the Discussion section of the paper.

Africanized Conspiracies: I found this section to be difficult to navigate between editorial narrative, theoretical discourse, quotes from focus groups/interviews, and attributions to other beliefs.

Societal Polarization: This section also has a combination of data, theory, literature review, and conjecture thrown together.

Conclusion: There is too much editorializing in this section. I believe that there are some excellent objective findings from the qualitative data but it seems to be lost in editorial comments that do not provide any organized understanding or suggested ways forward.

Round 2

Reviewer 3 Report

Thank you for your thoughtful and thorough responses to the comments and suggestions in the original review. I believe that your modifications have greatly improved the presentation of your research and adds value to the overall study of vaccine hesitancy during the COVID-19 pandemic.